# Left-to-Left Acquired Cardiac Shunt: Aorto-Left Atrial Fistula Due to Aortic Infective Endocarditis in a Dog

**DOI:** 10.3390/ani14172451

**Published:** 2024-08-23

**Authors:** Teodora Popa, Cosmin Petru Peștean, Irina Constantin, Alexandra Cofaru, Raluca Murariu, Flaviu-Alexandru Tăbăran, Iuliu Călin Scurtu

**Affiliations:** 1Department of Small Animal Internal Medicine, University of Agricultural Sciences and Veterinary Medicine, 400372 Cluj-Napoca, Romania; dorrapopa@gmail.com (T.P.); alexandra.cofaru@usamvcluj.ro (A.C.); iuliu.scurtu@usamvcluj.ro (I.C.S.); 2Department of Surgical Techniques and Propaedeutics, University of Agricultural Sciences and Veterinary Medicine, 400372 Cluj-Napoca, Romania; cosmin.pestean@usamvcluj.ro; 3Department of Pathology, University of Agricultural Sciences and Veterinary Medicine, 400372 Cluj-Napoca, Romania; irina.constantin@usamvcluj.ro (I.C.); alexandru.tabaran@usamvcluj.ro (F.-A.T.)

**Keywords:** infective endocarditis, aortocardiac fistula, cardiac shunt, paravalvular abscess

## Abstract

**Simple Summary:**

Infective endocarditis represents a rare, acquired condition in animals, but also in humans, that typically involves one of the heart valves and mainly affecting the mitral and aortic valves in dogs. This report describes the clinical signs, the diagnostic approach and the postmortem investigations of a case confirmed with infective endocarditis at the level of the aortic valve but also the presence of a main consequence of the disease, represented by the development of paravalvular abscesses and subsequent fistulous tracts between heart chambers.

**Abstract:**

Infective endocarditis is a severe but rarely diagnosed disease, characterized by the presence of bacterial infection at the level of the cardiac valves. Although the incidence of the disease is very low, the consequences are severe and the prognosis is very poor, outlining a high mortality rate among cases. The present report highlights the case of a 7-year-old dog presented with abrupt changes in the respiratory pattern, obtunded and in lateral recumbency. The physical examination of the patient revealed fever and a IV/VI systolic heart murmur, with the point of maximal intensity on the left hemithorax. Echocardiography identified hyperechoic and cavitary changes beneath the aortic valves and a retrograde turbulent jet originating in the left ventricle outflow tract communicating with the left atrium through a rupture in the aortomitral intervalvular wall. Because of very unstable hemodynamic changes, the dog suddenly died despite the initiation of intensive care supportive treatment, and the postmortem evaluation of the heart confirms the suspicion of infective aortic endocarditis with the development of a paravalvular abscess and an aorto-left atrial fistula.

## 1. Introduction

The presence of infection at the level of the heart’s structures, also described as the bacterial invasion of the endocardium and the endothelium of the cardiac valves, is one of the most uncommon diagnosed acquired heart diseases in dogs. The prevalence of the disease is low, usually not exceeding 1% of the cases presented for a cardiological examination [1,2]. Some reports mention a prevalence of up to 6.6% of cases presented to referral facilities [3]. For the disease to develop, a few predisposing factors’ presence is required. The most important factors are represented by the damage of cardiac valve, most frequently aortic or mitral, alongside bacteriemia [3]. Congenital defects such as subaortic stenosis in dogs are known to facilitate the embedding of microorganisms at the level of the valve [3,4,5]. Other defects identified in conjunction with infective endocarditis (IE) are represented by a bicuspid aortic valve [4]. The nidus of the bacteriemia are various and can be located at the level of the skin (pyoderma, chronic infected wounds), respiratory system (pneumonia), urogenital tract or gastrointestinal tract [3]. The most common causative agents isolated are *Staphylococcus* spp., *Streptococcus* spp., *E. coli* and *Bartonella* spp. [2,3].

The clinical signs usually encountered at the initial presentation are various and nonspecific. The most common complaints expressed are lethargy, inappetence, weight loss, respiratory difficulties, and locomotor problems (lameness, inability to move, joint pain and swellings) [2]. Laboratory findings are also not directly suggestive of IE: blood cell count analysis reveals leukocytosis in most cases, with neutrophilia and monocytosis; thrombocytopenia can also be seen alongside mild nonregenerative anemia. On serum biochemistry, the most common changes are hypoalbuminemia, elevated hepatic enzymes, azotemia and high creatinine levels [2,3].

The diagnosis of IE is complex; while specific criteria were developed in human medicine, known as the Duke criteria [6], more recent interpretations used in veterinary medicine were proposed, following the major and minor criteria met in the original version (Table 1) [1,3,7]. In dogs, a definite diagnosis is supported if the changes in the affected valve are histopathologically confirmed, or if two major criteria are met or if one major and two minor criteria can be identified [1,5]. In the absence of sufficient criteria that define a positive diagnosis, suspicion can still be raised based on the presence of one major and one minor criterion or three minor criteria [1].

Consequences of the IE can also be encountered, the most commonly identified ones being congestive heart failure, arrythmias, thromboembolic disease and circulating immune complexes [1,2,3]. Less frequently seen consequences are represented by the development intercavitary fistulas [4,8,9,10].

The aim of this report is to present in detail the medical conduit used to diagnose IE associated with the severe consequences of paravalvular abscess development and fistula formation between the aorta and left atrial chamber in a dog.

**Table 1 animals-14-02451-t001:** Suggested criteria for diagnosis of infective endocarditis in dogs [1,3,6,11].

Major Criteria	Minor Criteria
Positive echocardiogram Vegetative lesions Erosive lesion Abscess New valvular insufficiency Mild aortic insufficiency in absence of subaortic stenosis or annuloaortic ectasia Positive blood culture	Fever Newly identified heart murmur Predisposing cardiac disease Subaortic stenosis Medium to large dog (>15 kg) Thromboembolic disease Immune mediated disease

## 2. Detailed Case Description

### 2.1. History and Clinical Examination

A 7-year-7-month-old, 50 kg, intact male American Staffordshire mix with a sudden onset of rapid, shallow breathing, impossibility of maintaining quadrupedal position and an incidentally newfound heart murmur upon clinical examination was referred to the Cardiology Department of the University of Agricultural Sciences and Veterinary Medicine of Cluj-Napoca (Romania), to have a detailed cardiological evaluation for further investigation of the heart murmur described.

The physical examination identifies a non-ambulatory patient, in permanent lateral recumbency, obtunded, fever (41 °C), tachypnea at 68 respirations per minute (rpm) and a heart rate of 115 beats per minute (bpm). Thoracic auscultation revealed a IV/VI holosystolic heart murmur with the point of maximal intensity on the left hemithorax, at the level of the mitral valve. Prior to the cardiological evaluation, a CBC count (Abaxis VetScan HM5 hematology analyzer, Abaxis Inc., Union City, CA, USA) and serum biochemistry analysis (Automatic Veterinary Chemistry Analyzer Element RC, Scil Animal Care Company, Alfort, France) were performed. The remarkable changes identified were neutrophilia of 14.61 × 10^9^/L (reference values, 3.00 to 12.00 × 10^9^/L) and mild lymphopenia of 0.62 × 10^9^/L (reference values, 1.00 to 4.80 × 10^9^/L). The serum biochemistry analysis showed mild hypoproteinemia of 4.9 g/dL (reference range, 5.5–7.5 g/dL) with mild hypoalbuminemia of 2.2 g/dL (reference range, 2.6–4.0 g/dL), a high alkaline phosphatase of 293 U/L (reference range, 11–101 U/L) and mild hypoglycemia of 58 mg/dL (reference range, 62–108 mg/dL). Because of the unknown status of parasite prevention in the patient, a multi-strip rapid test for *D. immitis* antigen, *A. phagocytophilum/platys*, *E. canis* and *B. burgdorferi* antibodies was performed (Idexx SNAP 4Dx Plus Test). The results were negative.

### 2.2. Diagnostic Imaging

Transthoracic echocardiography (TTE) was performed including real-time two-dimensional (2D), M-mode and Doppler investigations (Esaote MyLabX8 Vet unit, equipped with a dedicated phased array probe for large dogs, P1–5, Esaote SpA, Genova, Italy). The cardiac ultrasound was performed with continuous electrocardiographic monitoring. The electrocardiogram (ECG) showed a normal sinus rhythm with a ventricular rate of 112–115 bpm.

In the right parasternal long-axis view, the left ventricle (LV) appeared subjectively mildly dilated. Left ventricular dimensions were measured from the right parasternal short-axis view, at the level of papillary muscles, using M-mode. The measurements were compared to the reference intervals calculated using the equation derived by Cornell al. 2004, considering the 95th percentile and were as follows: the left ventricular internal dimension in diastole (LVIDd) normalized (LVIDdN) to the patient’s body weight using the allometric formula [12]
LVIDdN = LVIDd (cm)/weight (kg)^0.294^
was 1.81 (1.35–1.73) which was slightly increased, while in systole (LVIDs), the normalized value (LVIDdN) was normal at 1.1 (0.79–1.14), calculated using the formula [12]
LVIDSN = LVIDs (cm)/weight (kg)^0.315^

Normal values were also obtained by measuring the end-diastolic interventricular septum (IVSd), whose normalized value (IVSdN) measured 0.38 (0.33–0.52), while in systole (IVSs), the normalized value (IVSsN) was 0.63 (0.43–0.78); at the end-diastolic left ventricular posterior wall (LVPWd), the normalized value (LVPWdN) measured, 0.45 (0.28–0.6), was normal, and during systole (LVPWs), the normalized value (LVPWsN) measured, 0.65 (0.46–0.88), was also normal.

The normalized values were obtained by using the allometric formulas [12]:
IVSdN = IVSd (cm)/weight (kg)^0.241^

IVSsN = IVSd (cm)/weight (kg)^0.241^

LVPWdN = LVPWd (cm)/weight (kg)^0.232^

LVPWsN = LVPWs (cm)/weight (kg)^0.222^


The left ventricular systolic function was preserved, with a normal fractional shortening of 36%. The short-axis view, at the base of the heart, shows a normal LA/Ao ratio of 1.29. Moderate B lines were identified on both sides of the chest.

The right chambers of the heart showed no remarkable changes. The investigation of the aortic valves and trunk showed notable changes in the aortic cusps. Immediately under the non-coronary aortic cusp, isoechoic visible oscillating irregular-shaped growths were seen attached to the mitral–aortic intervalvular wall, which outline a rounded, double-cavitary “aneurysmal-like” structure (Figure 1a, Appendix A) that resembles a paravalvular abscess. The mitral–aortic wall shows a discontinuation, at the level of which the isoechoic structures previously described are seen bulging into the left atrium (LA) (Figure 1b,c; Appendix A). In the short-axis view at the base of the heart, the changes are confirmed to be affecting the non-coronary aortic cusp, where cavitary changes are obviously seen (Figure 1d).

Color Doppler interrogation shows a continuous systolic–diastolic turbulent flow entering the LA at the level of the modified aortic wall (Figure 2a,b; Appendix A). Continuous-wave Doppler investigation of the turbulent flow, retrograde in the LA, shows a peak systolic flow velocity of 5.56 m/s, a value corresponding to a pressure gradient of 123 mmHg, confirming the systemic origin of the flow. The peak velocity of the diastolic component of the non-laminar flow was 3.49 m/s (pressure gradient 49 mmHg). Both Doppler investigations confirm the presence of communication between the aorta and the LA. As the passage of the turbulent flow is seen both in systole and in diastole (Figure 2a,b; Appendix A), it is believed that the size of the defect is extending distally from the LV outflow tract and involving the sinus of Valsalva.

Other echocardiographic changes include a mild tricuspid regurgitation. The peak velocity of the aortic flow was 2.5 m/s (pressure gradient 25 mmHg), suggesting that abnormal structure induced a mild subaortic stenosis. Pulsed-wave Doppler interrogation of the early diastolic transmitral flow (E wave) shows a peak velocity of 1.8 m/s, a value that suggests a high left ventricular filling pressure [13], with a late diastolic flow (A wave) of 0.64 m/s .

The final suspected diagnosis was aortic IE with ruptured paravalvular abscess and fistula formation towards the LA. Due to the significant echocardiographic changes found and the clinical status of the patient, no other diagnostic imaging techniques were performed at the time.

Intensive care treatment was initiated, consisting of oxygen therapy, IV butorphanol (0.2 mg/kg), IV furosemide (2 mg/kg) and IV antibiotics: amoxicillin/clavulanic acid (Amoxiplus^®^ (Antibiotice S.A., Iași, România) 1000 mg/200 mg (off-label), 15 mg/kg). In spite of initial efforts, the patient died shortly after. The owner only agreed to the postmortem evaluation of the heart and allowed sampling for further studies. The cause of death was established as acute heart failure with development of fulminant pulmonary edema due to the ruptured aortic wall. Other comorbidities could not be excluded.

### 2.3. Histopathological Findings

The dissection of the heart was carried out following the “inflow–outflow method” as previously described [14].

On gross examination, at the level of the aortic outflow tract, there was a focal, well-demarcated, oval fistula with anfractuous margins, measuring 1.3 × 0.7 cm (Figure 3a,c). The margins of the aortic–atrial fistula were markedly congested, and several parietal-adherent thrombi protruded within the aortic tract and LA. Focally, the inflammatory process extends until the deep area of the aortic cusps.

Histologically, the endocardium bordering the margins of the fistula was extensively ulcerated, with adherent septic thrombi consisting of fibrin admixed with necrotic cell debris, red blood cells, degenerated neutrophils and many basophilic colonies of cocci measuring 1–2 µm (Figure 3b,d). Extensively, the inflammation, consisting mainly of neutrophils admixed with fewer macrophages, infiltrates the subjacent myocardium and the subepicardial fat.

## 3. Discussion

Infective endocarditis in dogs is a rarely diagnosed affection that is described as the microbial invasion of the endocardium and cardiac valves, the prognosis of which is poor [1,2,3]. The most common affected sites are the aortic and mitral valve; the patients suffering from aortic IE hold a much worse prognosis, with a median survival time of 3 days [3]. The prevalence of the disease is less than 1%, although it is likely to be underestimated, and the mortality arises close to 78% [1]. In humans, a review involving 2371 cases from 15 different population types also revealed a low incidence of IE of 1.4 to 6.2 cases over 100,000 persons [15].

The pathogenesis of IE is poorly understood. It has been described as mainly consisting of an infected platelet and fibrin clot that is clinging onto the cardiac valves [16]. Normally, the endothelium and the surface of the cardiac valves are naturally resistant to bacterial invasion. Underlying valve diseases such as subaortic stenosis [3], or more recently described in a dog, bicuspid aortic valve [2], lead to turbulent flows that damage the endothelium and also create mechanical stress that makes the valve vulnerable to infections [16].

Valvular infections are always preceded by bacteriemia, which instantly trigger the involvement of the coagulation system. Bacteria are quickly recognized by the endothelial cells, leukocytes and platelets, triggering the extrinsic and intrinsic coagulation pathways. The coagulation cascade leads to activation of thrombin, necessary for the process of hemostasis, coagulation and inflammation and will further initiate the activation of platelets and fibrin formation. A sterile thrombotic vegetation is formed initially consisting mainly of platelets and fibrin which will provide the substrate for bacterial adhesion and entrapment. The circulating bacteria will then adhere directly or indirectly with the help of platelets to the damaged or inflamed cardiac endothelium, being shielded in the fibrin layer formed from the constant attack by the immune system [16]. The adherence is promoted by microbial surface components; the most potent ones are represented by bacteria from the genera *Staphylococcus* and *Streptococcus* [3].

Although very rare, one life-threatening consequence of IE is paravalvular abscess formation, whose rupture can represent the first initial step in developing an aortocardiac fistula (ACF). With IE, the infection is most likely easier to spread in an area with lack of vascularization; the avascular junctional area between the mitral and aortic annulus is often targeted, the result being abscess formation and rupture into one of the chambers of the heart, this way forming an ACF [17].

Fistulous tracts between the aorta and a heart chamber are uncommon [18] but have been reported in humans [17,18,19,20,21,22,23,24], dogs [4,5,8,9,25,26], one cat [10] and horses [27,28].

In humans, retrospective studies show an incidence of <2% of acquired ACFs [19], the majority of cases being secondary to IE (~25%) [21], other causes being represented by aortic aneurysms (22.1%) or aortic dissection; primary congenital causes (11.8%) are also reported, associated with genetic and connective tissue disorders [17,22]. The most common communication between the aorta and a heart chamber occurs in the right atrium, right ventricle and less common into the LA [17,22].

In dogs, typically, the defects follow the aorto-right atrium path, also known as Gerbode defects, which could be primary or also secondary to IE [5,25,26]. More recent case reports suggest the possibility of a fistula formation between the left ventricle outflow tract and LA consecutive to the development of IE [4,8,9].

Although IE is extremely rare in cats [3], the pathology has recently been described in association with an underlying condition such as valvular aortic stenosis [10]. Mechanical valvular lesions are known to be a predisposing factor for the occurrence of IE in dogs (subaortic stenosis) [1,3] but it is the first case described in cats as such. The case reported showed an even more rare complication, a secondary fistula originating from the left coronary sinus of Valsalva to the LA [10].

An increased number of ACFs have been reported in middle-aged horses, the congenital form being the most commonly encountered in their case, where an incomplete fusion between the annulus fibrosus of the aortic valve and the tunica media of the aorta is described. This discontinuity between layers causes progressive dilation of the weakened area, until it ruptures, leading to formation of tracts originating most commonly in the right aortic sinus that dissect through the aortic ring to the right ventricle, right atrium or in the myocardial septum [27,28].

The history and clinical signs include nonspecific symptoms of extracardiac systemic illness, including lethargy, weakness, lameness, respiratory abnormalities and collapse [3]. Fever and a newly identified cardiac murmur that could change in intensity throughout the progression of the pathology could raise suspicion of the disease, even though in one study, only 41% of the dogs diagnosed with IE were associated with a new heart murmur [29].

In our case, the clinical signs noted designed the pathway to the suspicion of IE (fever, collapse, heart murmur); however, previous history was lacking. A few other similar cases are reported in small animal medicine, cases that confirm the variability and nonspecificity of clinical signs. One of the dogs recorded with IE and ACF development showed no relatable signs towards this diagnosis other than a newly diagnosed systo-diastolic heart murmur, incidentally found as he was presented for a vomiting episode [4]. Another dog with the same issue initially presented for anorexia, weakness and progressive lameness. The clinical examination revealed fever, a systolic heart murmur, joint effusions along the posterior limbs and mouth ulcerations and malodor [9].

Because the affection has a rapid progression and is very aggressive, signs of acute heart failure can be seen, with respiratory distress caused by fulminant pulmonary edema [30].

The results of the CBC count revealed neutrophilia and lymphopenia. Mild changes showed up in the serum biochemistry: hypoproteinemia with hypoalbuminemia, high ALP and mild hypoglycemia. Blood culture was not available for this case nor PCR investigation. Similarities to these blood test results were noted in another dog that presented the same changes alongside thrombocytopenia, increased renal markers (BUN, creatinine), high cholesterol and triglyceride levels [4], while another showed all normal values except hypoglycemia and thrombocytopenia [10].

Complementary diagnosis methods included echocardiography which revealed the most significant changes. A LV diastolic dilation was detected, with no apparent changes over the right side of the heart. Remarkable structural changes were noted at the level of the aortic valve, where a paravalvular abscess was identified. Color and spectral Doppler interrogation revealed a continuous turbulent flow through the ruptured abscess, passing from the aorta to the LA. This showed an inconsistency with the clinical examination where only a systolic murmur was audible. Having the echocardiographic confirmation, we consider the auscultation as limited due to a few factors: the clinical state of the patient (permanent lateral recumbency) and tachypnea.

The clinical presentation of the patient was suspected to be secondary to acute heart failure. We believe that the rapid development of the events was caused by the rupture of the aorto-left atrial wall that was obviously damaged by the infection. The hemodynamic changes caused by the large regurgitant volume into the LA were severe, as the chamber itself did not have time to adapt to such change. This led to a massive increase in the LA pressure (confirmed by the high velocity of early diastolic transmitral flow). Based on the clinical signs and echocardiographic changes found (mitral valve E wave 1.8 m/s), pulmonary edema was suspected, confirmed afterwards with the presence of B lines on chest ultrasound. However, other diagnostic methods were not used.

The presumptive diagnosis of IE in this case was reached following the proposed modified Duke criteria (Table 1) [1,3,6,11], known in human medicine and previously described in dogs. In the presented case, antemortem diagnosis met the major criteria of echocardiography and a few minor predisposing and clinical criteria (large breed, heart murmur, and fever) [6,11]. An echocardiographic examination in this case showed remarkable changes (the aortic valve appearance, the identification of a paravalvular abscess, the turbulent flow that confirmed the rupture of the abscess and its communication with the LA) that are consistent with IE. The histopathological findings on the heart valves supported the definite diagnosis of IE, with abscess formation and development of an aorto-left atrial fistula.

## 4. Conclusions

In conclusion, this case report describes one the rarest and most life-threatening consequences of an aggressive disease and gives key changes that could ease the diagnosis of such disease. Although an aorto-left atrial fistula was described before and in more recent publications, the identification of a paravalvular abscess is highlighted for the first time in dogs.

## Figures and Tables

**Figure 1 animals-14-02451-f001:**
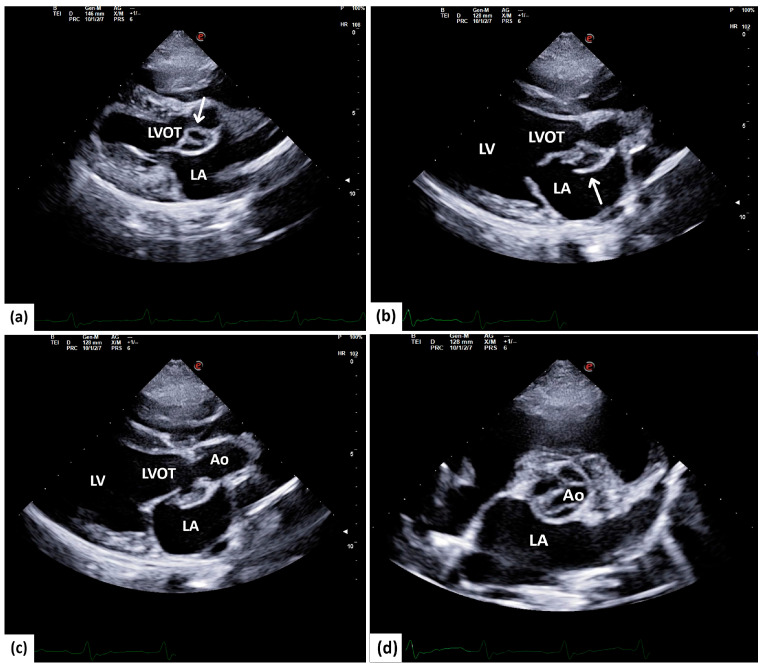
A five-chamber long-axis view in a 7.7-year-old American Staffordshire mix male. (**a**) A double-cavitary structure with isoechoic walls is identified at the level of the left ventricle outflow tract (arrow). (**b**,**c**) Discontinuation of the aortomitral intervalvular wall (arrow) with an isoechoic structure protrusion through the discontinuation into the left atrium (LA). (**d**) A short-axis view at the base of the heart shows cavitary changes on the non-coronary aortic cusp. LVOT = left ventricle outflow tract, LA = left atrium, LV = left ventricle, AO = aorta.

**Figure 2 animals-14-02451-f002:**
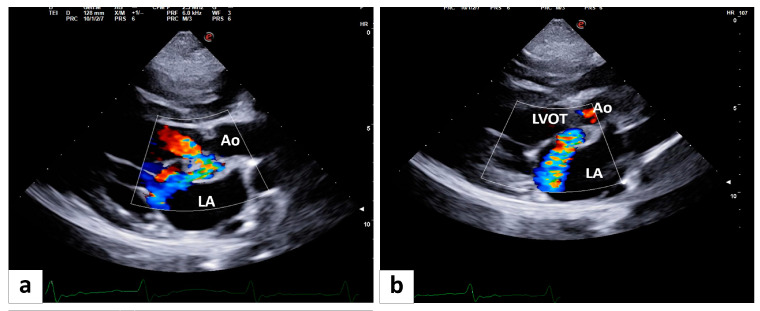
A long-axis five-chamber view in a 7.7-year-old American Staffordshire mix male. Color Doppler interrogation at the level of the defect underlined a continuous turbulent flow both in end-diastole (**a**) and end-systole (**b**).

**Figure 3 animals-14-02451-f003:**
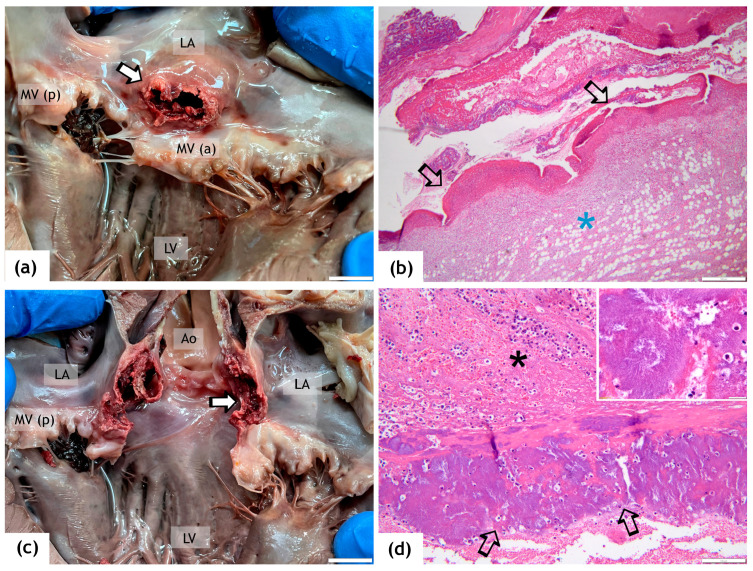
Gross and histopathological images of the aortic–atrial fistula. (**a**,**c**) Focally, the left atrial wall is disrupted by an anfractuous fistula (arrows) located within the aortic outflow tract and near the insertion of the anterior leaflet of the mitral valve. (**b**,**d**) The margins of the fistula are extensively ulcerated (black arrows) and covered with septic thrombi containing many basophilic colonies of cocci ((**d**) and inset) admixed with fibrin, blood, necrotic cell debris and neutrophils (black asterisk); the inflammatory process extensively infiltrate and replace the subjacent myocardium and subepicardial fat (blue asterisk). H&E stain, ob×10 (**b**) and ×40 (**d**). Scale bar = 1 cm for (**a**,**c**), 400 µm for (**b**), 100 µm for (**d**) and 20 µm for the inset. LV = left ventricle, LA = left atrium, MV = mitral valve, MV (a) = anterior leaflet of the mitral valve, MV (p) = posterior leaflet of the mitral valve, Ao = aorta.

## Data Availability

Data are contained within the article and Appendix A.

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
