# Peer review of "Left-to-Left Acquired Cardiac Shunt: Aorto-Left Atrial Fistula Due to Aortic Infective Endocarditis in a Dog"

_animals, 2024, doi:10.3390/ani14172451_

Round 1
Reviewer 1 Report
Comments and Suggestions for Authors
Very interesting case! Overall, the paper is well thought out with a good review of IE with well described diagnostics and postmortem evaluation. I would suggest culture/PCR/ID of the bacterial agent, if possible, to better meet the criteria mentioned in the paper. Rewording of a few sentences will aid in the flow and aid the reader in understanding the main points. The wording of the owner permission should be clarified to avoid a negative view of the work performed.
A few comments:
- I would take care regarding using the term "rarest" unless you have significant statistics for this disease. One of your references (1 Macdonald) suggests it is common while the others suggest rare so I would verify the accuracy of the statement.
-Within the abstract the tense changes between line 26 and 27.
- During the introduction it would be helpful to discuss the criteria and refer to table 1.
- Line 51 "the" is unnecessary.
- Line 110 discussion about the owner not allowing necropsy should be reworded. Perhaps say owner elected for postmortem evaluation of the heart, rather than they did not agree with necropsy (which was at least partially performed to retrieve the heart). Along this line I would also want to know the cause of death. Was this not determined due to the limits of necropsy?
-Line 124 Is "Discretion" of the heart meant to be "dissection"?
-Line 133 I would suggest you add bacterial culture or PCR results.
- Line 181-183 this sentence is confusing and may need rewording.
- Line 190 "arise" should be changed to "raise" or "arouse".
- Lines 195-199 Seem like they should be in the introduction as that is where I wanted to learn about the criteria. Then in discussion mention if the case met the criteria.
- Line 199-203 This sentence should be reworded.
-Line 229 This reference says IE is common.
-Line 235 This reference, if used for statistics refers to a paper by Sykes for the original statistics. I would add or use it instead of a secondary reference.
Comments on the Quality of English LanguageSee above comments.
Author Response
Thank you so much for reading and taking time to review our paper!
1: - I would take care regarding using the term "rarest" unless you have significant statistics for this disease. One of your references (1 Macdonald) suggests it is common while the others suggest rare so I would verify the accuracy of the statement.
Thank you! The term was used based on a conclusion of all our references. We just revised this issue and MacDonald's paper - I agree that it's stated in the introduction that the "diagnosed" cases have a low prevalence and mentioned the true prevalence of undiagnosed cases as it indeed might be higher. However the author then supports the rarity in the "Summary" section where she states "IE is an uncommon, deadly, and elusive disease to diagnose in dogs".
2: -Within the abstract the tense changes between line 26 and 27.
Thank you for pointing this out, we definetely missed that! I have changed it to the same tense (marked in red - line 27)
3: - During the introduction it would be helpful to discuss the criteria and refer to table 1.
Agreed. We have changed the structure accordingly - the table and information about the definite diagnosis/possible diagnosis are now in the introduction paragraph (lines 62-66 text marked in red, table 1 now moved (line 75).
4: - Line 51 "the" is unnecessary.
Thank you for pointing this out!
5: - Line 110 discussion about the owner not allowing necropsy should be reworded. Perhaps say owner elected for postmortem evaluation of the heart, rather than they did not agree with necropsy (which was at least partially performed to retrieve the heart). Along this line I would also want to know the cause of death. Was this not determined due to the limits of necropsy?
Thank you for this suggestion, much appreciated. We do agree it needs rephrasing - have accordingly changed the text and is now marked in red through lines 153-157
6: -Line 124 Is "Discretion" of the heart meant to be "dissection"?
Oh thanks for pointing this out, definitely meant dissection! Marked and now can be found in line 169
7: -Line 133 I would suggest you add bacterial culture or PCR results.
Thank you! Unfortunately we did not have access to these types of tests.
8: - Line 181-183 this sentence is confusing and may need rewording.
Thank you! We have reviewed and slightly reworded it. The changes are marked in red between lines 239-242.
9: - Line 190 "arise" should be changed to "raise" or "arouse".
I agree with this comment, I have changed it accordingly and switched the term to "raise" as you kindly suggested. Line 249
10: - Lines 195-199 Seem like they should be in the introduction as that is where I wanted to learn about the criteria. Then in discussion mention if the case met the criteria.
Thank you for suggesting this! I have changed as I was reviewing it with the previous comment and now rephrased a little in the discussion section.
11: - Line 199-203 This sentence should be reworded.
Thank you! I agree it was very hard to read and to follow. I changed the content now at lines 283-286
12: - Line 229 This reference says IE is common.
Thank you! In the "Summary" section the author states "IE is an uncommon, deadly, and elusive disease to diagnose in dogs".
13: - -Line 235 This reference, if used for statistics refers to a paper by Sykes for the original statistics. I would add or use it instead of a secondary reference.
Thank you for pointing this out! This is an article with a lot of helpful information to support our report. I have added it as a primary reference and citations can be found in the introduction part.
Reviewer 2 Report
Comments and Suggestions for Authors
The manuscript titled "Left-to-left acquired cardiac shunt: aorto-left atrial fistula due to aortic infective endocarditis in a dog" describes a case of pathological communication between the left ventricular outflow tract and the left atrium secondary to endocarditis. The clinical section is very lacking in the description of echocardiographic findings and the pathophysiology of the observed alteration. Below are some comments:
Lines 36-38: please provide references about the prevalence of IE in dogs.
Lines 41-42: please provide references
Lines 53-55: please provide references
Line 69: Has the cause of the tachypnea been investigated? Was the dog in congestive heart failure? Were chest X-rays taken?
Lines70-71: In lines 99-100 authors stated “Color Doppler interrogation shows a continuous systolic-diastolic turbulent flow entering the left atrium at the level of the modified aortic wall”. Being the turbulent flow continuous, why the murmur was systolic? Do you have an hypothesis of this inconsistence? Please report it in the discussion section.
Lines 71-80: please provide the names of cell blood counter, biochemical analysis instrument and multi strip rapid test, performed to the dog.
Lines 82-83: please provide the name of the ultrasonographic unit and probe used for the echocardiographic examination.
Lines 83-90: Please provide all the relevant echocardiographic variables measured at list the LA/Ao diameter, the indices of LV systolic function (at least FS and EF), the thickness of the LV walls, the normalized LVID in systole, the transmitral flow (E and A) velocity, the TDI-derived E1 and A1 values. What about the ECG?
Line 91: Please specify which parameters were within the normal limits.
Line 102: please provide the diastolic velocity and relative pressure gradient.
Line 105: please report the tricuspid regurgitation velocity.
Line 110: please specify “the intensive care treatment” performed to the dog and why.
DISCUSSION
Lines 152-153: please provide references.
Line 176: In cats?
The discussion section should thoroughly discuss the clinical and instrumental findings, comparing them with previous publications in human and veterinary medicine that describe similar clinical cases. Additionally, the discussion should delve into the pathophysiology of the observed alterations and the diagnostic limitations of the manuscript.
Comments on the Quality of English LanguageThe manuscript titled "Left-to-left acquired cardiac shunt: aorto-left atrial fistula due to aortic infective endocarditis in a dog" describes a case of pathological communication between the left ventricular outflow tract and the left atrium secondary to endocarditis. The clinical section is very lacking in the description of echocardiographic findings and the pathophysiology of the observed alteration. Below are some comments:
Lines 36-38: please provide references about the prevalence of IE in dogs.
Lines 41-42: please provide references
Lines 53-55: please provide references
Line 69: Has the cause of the tachypnea been investigated? Was the dog in congestive heart failure? Were chest X-rays taken?
Lines70-71: In lines 99-100 authors stated “Color Doppler interrogation shows a continuous systolic-diastolic turbulent flow entering the left atrium at the level of the modified aortic wall”. Being the turbulent flow continuous, why the murmur was systolic? Do you have an hypothesis of this inconsistence? Please report it in the discussion section.
Lines 71-80: please provide the names of cell blood counter, biochemical analysis instrument and multi strip rapid test, performed to the dog.
Lines 82-83: please provide the name of the ultrasonographic unit and probe used for the echocardiographic examination.
Lines 83-90: Please provide all the relevant echocardiographic variables measured at list the LA/Ao diameter, the indices of LV systolic function (at least FS and EF), the thickness of the LV walls, the normalized LVID in systole, the transmitral flow (E and A) velocity, the TDI-derived E1 and A1 values. What about the ECG?
Line 91: Please specify which parameters were within the normal limits.
Line 102: please provide the diastolic velocity and relative pressure gradient.
Line 105: please report the tricuspid regurgitation velocity.
Line 110: please specify “the intensive care treatment” performed to the dog and why.
DISCUSSION
Lines 152-153: please provide references.
Line 176: In cats?
The discussion section should thoroughly discuss the clinical and instrumental findings, comparing them with previous publications in human and veterinary medicine that describe similar clinical cases. Additionally, the discussion should delve into the pathophysiology of the observed alterations and the diagnostic limitations of the manuscript.
Author Response
Thank you for reviewing our paper!
Please accept our responses:
1: - Lines 36-38: please provide references about the prevalence of IE in dogs.
Thank you! I reviewed the lines and accordingly changed the content - Lines 38-41 (marked in red) now contain references + added in an extra sentence about the prevalence signaled in the references.
2: - Lines 41-42: please provide references
Thank you! Citations are now provided in the document - line 45.
3: - Lines 53-55: please provide references (now lines 69-70)
Thank you - references are now added in lines 69-70 - also the sentence has been reviewed and rephrased based on the references.
4: - Line 69: Has the cause of the tachypnea been investigated? Was the dog in congestive heart failure? Were chest X-rays taken?
Thank you for this comment.
The cause of the tachypnea was suspected to be a combination of stress (a patient in lateral recumbency not able to move), fever, pulmonary edema (B lines) and potential other comorbidities.
The first complementary examination performed besides bloodwork was the cardiological exam due to the heart murmur found. Based on the major echocardiographic changes, the suspicion of disease and the instability of the patient the focus switched towards trying to stabilize the patient and no other exams were performed. Unfortunately, the patient died very shortly after the cardiological examination, so chest X Rays were not taken.
I revised and now the limitation of other diagnostic complementary exams is added to the text. Marked in red - LINE 148-150
5: - Lines70-71: In lines 99-100 authors stated “Color Doppler interrogation shows a continuous systolic-diastolic turbulent flow entering the left atrium at the level of the modified aortic wall”. Being the turbulent flow continuous, why the murmur was systolic? Do you have an hypothesis of this inconsistence? Please report it in the discussion section.
Thanks for pointing this out. We think the auscultation was limited first by a few clinical factors - permanent lateral recumbency on the right side, a slightly increased heart rate for the size of the patient and tachypnea. We confirmed the continuous flow then by Color Doppler, and also we think the diastolic component was missed during our heart auscultation.
I reviewed the discussion section and added this to it. Thank you. Marked in red - lines 275-278
6: - Lines 71-80: please provide the names of cell blood counter, biochemical analysis instrument and multi strip rapid test, performed to the dog.
Thank you! Rephrased slightly and I added them to the manuscript - marked in red - LINES 90-92 + line 100
7: -Lines 82-83: please provide the name of the ultrasonographic unit and probe used for the echocardiographic examination.
Thank you! Rephrased slightly and I added them to the manuscript - marked in red - LINE 104-105
8: -Lines 83-90: Please provide all the relevant echocardiographic variables measured at list the LA/Ao diameter, the indices of LV systolic function (at least FS and EF), the thickness of the LV walls, the normalized LVID in systole, the transmitral flow (E and A) velocity, the TDI-derived E1 and A1 values. What about the ECG?
We agree. We reviewed all the echocardiographic variables and all measurements are now listed in the appropriate section. We do not use TDI routinely in dogs so can not provide data for this parameter. Thank you for pointing this out. All marked in red, lines 111-126
9: - Line 91: Please specify which parameters were within the normal limits.
Thank you! They have all been reviewed and marked lines 111-126
10: -Line 102: please provide the diastolic velocity and relative pressure gradient.
Agreed, added in the text, thank you! Lines 138-139
11: -Line 105: please report the tricuspid regurgitation velocity.
Thank you for pointing this out, unfortunately the regurgitation was very mild and it didn't allow an appropriate measurement.
12: - Line 110: please specify “the intensive care treatment” performed to the dog and why.
Thank you! Revised and added to the script. Lines 151-153
13: - Lines 152-153: please provide references.
Added in the text, thank you!
14: - Line 176: In cats?
Thank you!
Yes, the line should be about cats. It made a parallel of the case of IE and development of a fistulous tract in a cat with aortic stenosis, condition that was only observed as a predisposing factor in dogs before. I understand it sounded confusing.
I rephrased it accordingly.
- The discussion section should thoroughly discuss the clinical and instrumental findings, comparing them with previous publications in human and veterinary medicine that describe similar clinical cases. Additionally, the discussion should delve into the pathophysiology of the observed alterations and the diagnostic limitations of the manuscript.
Thank you! The discussion section is now revised and contains a lot more information.
Round 2
Reviewer 2 Report
Comments and Suggestions for Authors
REVIEW 2
I thank the authors for making the requested changes. However, some fundamental pathophysiological aspects still need to be clarified, and the description of the procedures performed and the exact location of the lesion need to be improved. Moreover, authors should carefully read the "Instructions for Authors" of the journal “Animals” and apply them in the manuscript. (Eg. When defined for the first time, the acronym/abbreviation/initialism should be added in parentheses after the written-out form.)
Finally, I suggest the authors submit the manuscript to an English language editing service.
Below are the requested revisions:
ABSTRACT
Line 21: delete (IE)
Line 26: replace “heard best on the left hemithorax” with “point of maximal intensity on the left hemithorax”
INTRODUCTION:
Line 59: replace “infective endocarditis” with “IE”
Line 67: replace “infective endocarditis” with “IE”
Line 72: replace “infective endocarditis” with “IE”
Line 75: replace “IE” with “infective endocarditis”
Table 1: the font of the table should be the same as the text (Subaortic stenosis is not)
CASE DESCRIPTION
Line 84: indicate the country ….of Cluj-Napoca (Romania)
Line 108: add (bpm) after “beats per minute”
Line 109: replace “in the right parasternal section, long axis” with “in the right parasternal long axis view”
Lines 109-110: Since the LA/Ao ratio is 1.29, the LA is not dilated, and the sentence should ben“In the right parasternal section, long axis, the general aspect of the heart shows a mild dilation of both left heart chambers” should be deleted or replaced with 'in the right parasternal long axis view the left ventricle (LV) appeared subjectively mildly dilated'."
Line 111: please replace “In short axis, at the level of papillary muscles, using 2D measurements and M mode, the measurements were as follows:” with “Left ventricular (LV) dimension were measured from the right parasternal short axis view, at the level of papillary muscles, with M-mode and B-mode imaging”…
Lines 111-117: The authors should specify that the reference intervals were obtained using the formula by Cornell et al. 2004, considering the 95th percentile. In this way, they can correctly state the presence of a slight increase in LVIDdN 1.81 [reference 1.35-1.73], but not an increase in IVSdN, which is 0.38 (RR 0.33-0.52) neither an increase of the LVIDsN which is 1,1 (RR 0,79-1,14). Furthermore, it is not necessary to report the LV dimension in both the values, mm (LVIDd=57.1) and normalized values (LVIDdN=1.81); the authors should choose whether to report one or the other. In my opinion, it is better to report the normalized values.
Line 115: add “(IVSs)” after “IVS during systole”
Lines 115-116: add “(LVIDs)” after left ventricular chamber during systole
Line 111: The authors state that the LV measurements were obtained using B-mode and M-mode. Since the reference intervals calculated with the Cornell et al. 2004 formula refer to measurements obtained using M-mode, the authors should declare that the two methods are interchangeable, providing appropriate bibliographic references (e.g., 'Two-dimensional echocardiographic measures of left ventricular dimensions agree with M-mode measurements in dogs.' Rishniw et al. Journal of Veterinary Cardiology (2021)
LINE 133: please replace “left atrium” with “LA”
LINE 134: Was the defect located proximally (ventricular side) or distally (aortic side, Sinus of Valsalva) with respect to the aortic valve cusps? What do you exactly mean by the aortic outflow tract? If the shunt was between the LV outflow tract (left ventricular side) and the left atrium, it is assumed that the passage of a turbulent flow occurred solely and exclusively during systole. In this case authors should specify why in a communication between the LV outflow tract and LA there was a diastolic turbulent flow from the LV outflow tract and the LA. On the contrary, if the opening of the fistula was located distally to the aortic cusps (i.e., the Sinuses of Valsalva), the turbulent flow would have been continuous (systolic and diastolic) due to the pressure gradient. Please specify those aspects. If possible, provide more images or videos of the defect.
Line 135: please replace “left atrium” with “LA”
Line 136: please refer to the left atrium with the acronym LA
Line 173: please refer to the left atrium with the acronym LA
DISCUSSION
Line 268: Since the LA/Ao ratio is 1.29, the LA is not dilated, and the sentence “A very mild dilation of both left chambers was noted” should be replaced with “A LV diastolic dilation was detected”
Lines 290-292: a recent paper described a similar disease in a paper titled “Aorto-left atrial fistula secondary to aortic infective endocarditis in a dog with a bicuspid aortic valve “ Carrillo et al, Journal of Veterinary Cardiology (2024) 53, 13e19.
In the discussion section, the authors should explain to the reader the hypotheses that would explain the relationship between symptoms, echocardiographic findings, and pathological findings, referring to plausible hypotheses supported by adequate bibliographic references. In my opinion, in the clinical case you described, the presumed pulmonary edema was caused by the sudden rupture of the wall separating the LV outflow tract/aorta and LA, which was altered by the infection. This likely caused a rapid and intense increase in pressure in the left atrium and pulmonary veins, leading to edema and volume overload of the LV (similar to what happens in the case of sudden rupture of the chordae tendineae, see 'Acute mitral regurgitation with and without acute heart failure,' Dean Boudoulas et al. 2023 Heart Fail Rev.). The velocity and severity of the pressure increase could explain why the LA was not yet dilated, and why the E velocity was so high, and the sudden onset of clinical signs. Pulmonary edema was correctly suspected based on clinical symptoms, echocardiographic findings (E=1.8 m/s), and the presence of B-lines on chest ultrasound. However, the pulmonary edema was not confirmed by radiographic and pathological examinations because the reasons reported in the manuscript.
Comments on the Quality of English LanguageREVIEW 2
I thank the authors for making the requested changes. However, some fundamental pathophysiological aspects still need to be clarified, and the description of the procedures performed and the exact location of the lesion need to be improved. Moreover, authors should carefully read the "Instructions for Authors" of the journal “Animals” and apply them in the manuscript. (Eg. When defined for the first time, the acronym/abbreviation/initialism should be added in parentheses after the written-out form.)
Finally, I suggest the authors submit the manuscript to an English language editing service.
Below are the requested revisions:
ABSTRACT
Line 21: delete (IE)
Line 26: replace “heard best on the left hemithorax” with “point of maximal intensity on the left hemithorax”
INTRODUCTION:
Line 59: replace “infective endocarditis” with “IE”
Line 67: replace “infective endocarditis” with “IE”
Line 72: replace “infective endocarditis” with “IE”
Line 75: replace “IE” with “infective endocarditis”
Table 1: the font of the table should be the same as the text (Subaortic stenosis is not)
CASE DESCRIPTION
Line 84: indicate the country ….of Cluj-Napoca (Romania)
Line 108: add (bpm) after “beats per minute”
Line 109: replace “in the right parasternal section, long axis” with “in the right parasternal long axis view”
Lines 109-110: Since the LA/Ao ratio is 1.29, the LA is not dilated, and the sentence should ben“In the right parasternal section, long axis, the general aspect of the heart shows a mild dilation of both left heart chambers” should be deleted or replaced with 'in the right parasternal long axis view the left ventricle (LV) appeared subjectively mildly dilated'."
Line 111: please replace “In short axis, at the level of papillary muscles, using 2D measurements and M mode, the measurements were as follows:” with “Left ventricular (LV) dimension were measured from the right parasternal short axis view, at the level of papillary muscles, with M-mode and B-mode imaging”…
Lines 111-117: The authors should specify that the reference intervals were obtained using the formula by Cornell et al. 2004, considering the 95th percentile. In this way, they can correctly state the presence of a slight increase in LVIDdN 1.81 [reference 1.35-1.73], but not an increase in IVSdN, which is 0.38 (RR 0.33-0.52) neither an increase of the LVIDsN which is 1,1 (RR 0,79-1,14). Furthermore, it is not necessary to report the LV dimension in both the values, mm (LVIDd=57.1) and normalized values (LVIDdN=1.81); the authors should choose whether to report one or the other. In my opinion, it is better to report the normalized values.
Line 115: add “(IVSs)” after “IVS during systole”
Lines 115-116: add “(LVIDs)” after left ventricular chamber during systole
Line 111: The authors state that the LV measurements were obtained using B-mode and M-mode. Since the reference intervals calculated with the Cornell et al. 2004 formula refer to measurements obtained using M-mode, the authors should declare that the two methods are interchangeable, providing appropriate bibliographic references (e.g., 'Two-dimensional echocardiographic measures of left ventricular dimensions agree with M-mode measurements in dogs.' Rishniw et al. Journal of Veterinary Cardiology (2021)
LINE 133: please replace “left atrium” with “LA”
LINE 134: Was the defect located proximally (ventricular side) or distally (aortic side, Sinus of Valsalva) with respect to the aortic valve cusps? What do you exactly mean by the aortic outflow tract? If the shunt was between the LV outflow tract (left ventricular side) and the left atrium, it is assumed that the passage of a turbulent flow occurred solely and exclusively during systole. In this case authors should specify why in a communication between the LV outflow tract and LA there was a diastolic turbulent flow from the LV outflow tract and the LA. On the contrary, if the opening of the fistula was located distally to the aortic cusps (i.e., the Sinuses of Valsalva), the turbulent flow would have been continuous (systolic and diastolic) due to the pressure gradient. Please specify those aspects. If possible, provide more images or videos of the defect.
Line 135: please replace “left atrium” with “LA”
Line 136: please refer to the left atrium with the acronym LA
Line 173: please refer to the left atrium with the acronym LA
DISCUSSION
Line 268: Since the LA/Ao ratio is 1.29, the LA is not dilated, and the sentence “A very mild dilation of both left chambers was noted” should be replaced with “A LV diastolic dilation was detected”
Lines 290-292: a recent paper described a similar disease in a paper titled “Aorto-left atrial fistula secondary to aortic infective endocarditis in a dog with a bicuspid aortic valve “ Carrillo et al, Journal of Veterinary Cardiology (2024) 53, 13e19.
In the discussion section, the authors should explain to the reader the hypotheses that would explain the relationship between symptoms, echocardiographic findings, and pathological findings, referring to plausible hypotheses supported by adequate bibliographic references. In my opinion, in the clinical case you described, the presumed pulmonary edema was caused by the sudden rupture of the wall separating the LV outflow tract/aorta and LA, which was altered by the infection. This likely caused a rapid and intense increase in pressure in the left atrium and pulmonary veins, leading to edema and volume overload of the LV (similar to what happens in the case of sudden rupture of the chordae tendineae, see 'Acute mitral regurgitation with and without acute heart failure,' Dean Boudoulas et al. 2023 Heart Fail Rev.). The velocity and severity of the pressure increase could explain why the LA was not yet dilated, and why the E velocity was so high, and the sudden onset of clinical signs. Pulmonary edema was correctly suspected based on clinical symptoms, echocardiographic findings (E=1.8 m/s), and the presence of B-lines on chest ultrasound. However, the pulmonary edema was not confirmed by radiographic and pathological examinations because the reasons reported in the manuscript.
Round 3
Reviewer 2 Report
Comments and Suggestions for Authors
I thank the authors for making the requested changes, and the supplementary video files which are of excellent quality.
Here are some minor revisions:
Line 73: add “in a dog” at the end of the sentence.
Line 108: please delete the sentence “No abnormalities were intercepted.”
Lines 155-156: “that suggests a high left ventricular filling pressure”, please provide references.
Line 162: in my country there are currently no licensed intravenous amoxicillin clavulanate
preparations for animals. Is there such a possibility in your country? (if yes please provide the name of the drug) or have you made off-label use of this antibiotic (if so, you must declare it).
Figure 2, line 180: I think that (a) is a diastolic frame (end diastole) and (b) is a systolic frame (end systole). Please verify.
Lines 284-285: please provide references
Line 292: The 4DX SNAP test was negative please specify the test or delete the sentence as you already report this information in lines 99-101.
I recommend an English linguistic revision before publication
Comments on the Quality of English Language
I recommend an English linguistic revision before publication
